# Relationship between General Jump Types and Spike Jump Performance in Elite Female and Male Volleyball Players

**Philip X. Fuchs** [1,2,*]**, Julia Mitteregger** [1]**, Dominik Hoelbling** [3,4]**, Hans-Joachim K. Menzel** [5]**, Jeffrey W. Bell** [6]**,
Serge P. von Duvillard** [1] **and Herbert Wagner** [1,*]

1   Department of Sport and Exercise Science, University of Salzburg, 5400 Salzburg, Austria;
    Julia.Mitteregger@stud.sbg.ac.at (J.M.); Serge.Duvillard@sbg.ac.at (S.P.v.D.)
2   Department of Human Sciences, Society and Health, University of Cassino e Lazio Meridionale,
    03043 Cassino, Italy
3   Centre for Sport Science and University Sports, University of Vienna, 1150 Vienna, Austria; bzsu@univie.ac.at
4   School of Engineering, RMIT University, Melbourne, VIC 3000, Australia
5   Departamento de Esportes, Universidade Federal de Minas Gerais, Belo Horizonte 31270-901, Brazil;
    des@eeffto.ufmg.br
6   Science Department, Southwest Minnesota State University, Marshall, MN 56258, USA; science@smsu.edu
*   Correspondence: philip.fuchs@sbg.ac.at (P.X.F.); herbert.wagner@sbg.ac.at (H.W.); Tel.: +43-680-8044-4887 (H.W.)

**Featured Application: This investigation attempts to ascertain the value of testing general jump types for the assessment of specific jump performance and provides improved models including the most frequently used jump types with additional variables to more precisely predict sport-specific jump performance.**

**Abstract:** In performance testing, it is well-established that general jump types like squat and countermovement jumps have great reliability, but the relationship with volleyball spike jumps is unclear. The objectives of this study were to analyze the relationship between general and spike jumps and to provide improved models for predicting spike jump height by general jump performance. Thirty female and male elite volleyball players performed general and spike jumps in a randomized order. Two AMTI force plates (2000 Hz) and 13 Vicon MX cameras (250 Hz) captured kinematic and kinetic data. Correlation and stepwise-forward regression analyses were conducted at $p < 0.05$. Simple regression models with general jump height as the only predictor for spike jumps revealed $0.52 \leq R^2 \leq 0.76$ for all general jumps in both sexes ($p < 0.05$). Alternative models including rate of force development and impulse improved predictions during squat jumps from $R^2 = 0.76$ to $R^2 = 0.92$ ($p < 0.05$) in females and from $R^2 = 0.61$ to $R^2 = 0.71$ ($p < 0.05$) in males, and during countermovement jumps with arm swing from $R^2 = 0.52$ to $R^2 = 0.78$ ($p < 0.01$) in males. The findings include improved prediction models for spike jump height based on general jump performance. The derived formulas can be applied in general jump testing to improve the assessment of sport-specific spike jump performance.

**Keywords:** kinematic; kinetic; 3D analyses; influence; prediction; regression models; advanced performance testing; squat jump; countermovement jump

## 1. Introduction

Volleyball is a popular Olympic team sport, played in most countries of the world. Indoor volleyball is played by two teams of six players each on the court. The goal is to score more points than the opponent and, thereby, to win sets and the match. Studies have shown that serves, spikes, and blocks are the most important performance factors in volleyball [1,2]. In this context, jump height during spiking is a major determinant of success in male and female volleyball [3–5].

In accordance with theoretical descriptions [3], a volleyball spike jump (VSJ) is introduced by a three- or four-step approach with a countermovement, and followed by

a foot-planting phase to transfer velocity from horizontal to vertical. Decreased knee and upper body incline angles allow for lowering the center of mass (CoM) and, thus, increasing the acceleration path before the take-off [6]. A dynamic arm swing supports an explosive push-off with both legs [7], which is expressed by great peaks in the rate of ground reaction force development (RFD). Due to its complexity, the VSJ requires technical and coordinative skills. Hence, VSJ performance is not only affected by strength and power but also by movement-specific, technical coordination skills [6].

The complexity of the VSJ movement can make standardized performance testing and training difficult. Therefore, less complex general jump types are commonly used for testing [8] and training [9]. The general jump types for these purposes include the squat jump (SJ), countermovement jump without arm swing (CMJ), and countermovement jump with arm swing (CMJA) [8]. Despite not being sport-specific, these jump types are frequently used in testing and training to assess and improve jump performance in volleyball players [8,9]. This is due to the practical advantages of standardized general jumps, which are easier to be implemented and interpreted due to great reliability and factorial validity of these jumps [10]. However, it is unclear how suitable these general jump types are for the assessment of the sport-specific jump performance during VSJ. Clearly, though, each general jump type incorporates different aspects that have been described as characteristics for the VSJ performance: Generating great power and RFD during the SJ [11], involving a stretch-shortening-cycle during the CMJ, and supporting an explosive push-off when adding the arm swing during the CMJA. Therefore, the various general jump types may not fully capture the determinants of success in VSJ requiring additional considerations to assess VSJ performance based on various general jumps. Such differentiated considerations would allow the practical usage of general jump tests with greater reliability while advancing the explanatory power of the results that relate to VSJ performance.

Using this type of assessment, considerations should include factors occurring in the general jumps that are known to affect performance in VSJ. In accordance with the literature, the following factors fulfill both requirements by expressing the characteristics of the VSJ and correlating with VSJ height: Vertical decrease of CoM [6,12], maximum torso incline angle [6], and angular velocities in knee extension [6,13] and in shoulder flexion [6,13,14] (only relevant for CMJA) during the countermovement, and the peak RFD during upwards acceleration [15]. These are characteristics that can be measured during performance of general jump types and can benefit assessment of VSJ performance.

Other consequences of the complexity and technical-coordinative requirements in VSJ are the differences that have been observed in biomechanical movement characteristics and performance between sexes [6]. Sex differences in VSJ were previously reported in the vertical decrease of CoM, torso incline angle, minimal knee angles, the asymmetric usage of arm swing, and lower limb extension [6]. Therefore, sex-specific analyses are recommended.

The objectives of the current study are (1) to investigate the differences and relationship between the jump heights of three general jumps (SJ, CMJ, and CMJA) and VSJ; (2) to identify the most appropriate factors in general jumps to assess VSJ performance; and (3) to provide practical models for enhanced assessment and reliable predication for both sexes of VSJ height. Considering that the height of various jump types will differ in their relationship with VSJ height and that general jump height alone does not allow for strong association with VSJ height, we hypothesized that VSJ height can be more reliably predicted based on general jumps when specific factors in addition to general jump height are included.

## 2. Materials and Methods

### 2.1. Participants

Fifteen male and 15 female indoor volleyball players competing at the highest national level in Austria participated in this study. The final position in the Fédération Internationale de Volleyball world ranking were comparable between the male (age: 22.7 ± 4.3 years; body height: 1.88 ± 0.06 m; body mass: 80.9 ± 6.7 kg; reach height: 2.43 ± 0.07 m; training experience: 10.1 ± 5.9 years; training hours per week: 10.9 ± 4.3 h) and female (age: 19.9 ± 3.5 years;

body height: 1.79 ± 0.06 m; body mass: 70.5 ± 11.0 kg; reach height: 2.27 ± 0.08 m; training experience: 8.4 ± 3.9 years; training hours per week: 11.5 ± 2.2 h) team. In agreement with the Declaration of Helsinki, the local research committee approved the investigation. All athletes were healthy, in good physical condition, and provided written consent to participate in this study. For participants under the age of 18 years, parental consent was obtained.

### 2.2. Study Design and Test Protocol

After a general, supervised warm-up (e.g., running, jumping, and arm movements), all athletes performed as many test trials as necessary to familiarize themselves with the upcoming tasks. During the test trials of VSJ, each athlete found the individual optimal position of a volleyball hanging stationary from the ceiling and the self-selected approach distance and angle [6]. All athletes were required to complete a blocked series of SJ, CMJ, CMJA, and VSJ in a random order, each block consisting of 10 jumps. To avoid fatigue, athletes were free to take a 1-min break between trials. All athletes were instructed to jump as high as possible on each attempt. VSJ were valid if each foot contacted only one of two ground reaction force plates. Hereby, the players were instructed to not focus on the force plates but instead on jump height and ball velocity.

### 2.3. Data Collection and Processing

Kinematic data were obtained by 12 Vicon MX-13 cameras (Vicon, Oxford Ltd., Yarnton, UK) via 51 reflective markers of 14 mm diameter at a measuring frequency of 250 Hz. A Cleveland Clinic Marker set (Motion Analysis Corp, Santa Rosa, CA, USA) with clusters on the lower limbs [16] was used (Figure 1). Data were managed via Nexus 1.8 software (Vicon, Oxford Metrics Ltd., Oxfordshire, UK) and filtered according to Woltring [17]. A full-body 3D-model, segments, joints, and calculations of all variables were conducted via Visual 3D software (C-Motion Inc., Rocksville, MD, USA). The sagittal change of angle between two adjacent segments represented the rotation in the corresponding joint (shoulder: upper arm-torso; knee: shank-thigh) [13]. Two separate AMTI force plates (120 × 60 cm; AMTI, Watertown, MA, USA) were placed parallel to each other. Each force plate collected ground reaction forces of one leg at 2000 Hz (FP1 for the dominant leg, FP2 for the non-dominant leg). A fourth-order low-pass Butterworth filter (50 Hz) was applied, and kinetic data were normalized to body weight.

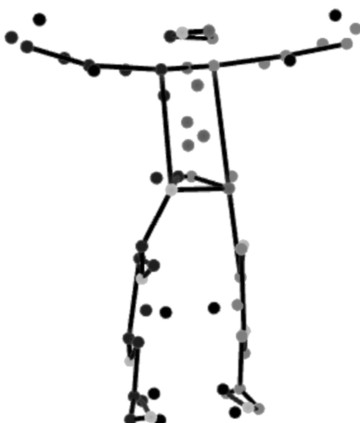

**Figure 1.** Marker locations of the Cleveland Clinic Marker set [16], including clusters on the lower limbs and calibration markers during static calibration of an athlete.

### 2.4. Variables and Definitions

Dominance of legs and arms was defined by the spiking arm for each individual player. Age, body height, body mass, reach height, training experience, and training hours per week were collected by an experienced researcher. Jump height was calculated via vertical CoM velocity at take-off. Vertical CoM displacement was defined as the difference

between the lowest vertical CoM position during jumping and CoM position in still stance, normalized by the position in still stance. Maximal angular velocities, RFD, and the vertical impulse on both force plates were obtained during the push-off phase. RFD was calculated for both legs separately and as the sum of both.

### 2.5. Statistical Analyses

Data (see Supplementary Materials) were analyzed via PASW Statistics 18.0 (SPSS Inc., Chicago, IL, USA) and visualized via Microsoft Excel Office 365 (Microsoft Corporation, Redmond, WA, USA). Normality of distribution was assessed via Shapiro-Wilk tests. Two-way analysis of variance (ANOVA) with repeated measures was conducted to determine main effects and interactions of jump height for jump types and sex. The effect size was expressed as partial eta square (pη2) and interpreted as small, medium, and large at the thresholds of 0.10, 0.25, and 0.40, respectively [18]. Separate male and female Pearson's Product Moment correlation coefficients ($r$) were calculated for the relationship between SJ, CMJ, CMJA, and VSJ heights as well as for the relationship between VSJ height and previously described kinematic and kinetic variables during general jump types. Subsequently, stepwise-forward regression models including adjusted $R^2$ were calculated to predict VSJ height based on kinematic and kinetic variables in addition to jump height of the general jump types. The significance level for all analyses was set a priori at $p < 0.05$. Post hoc power analyses were performed via G*Power 3.1.9.2 (Heinrich Heine University, Düsseldorf, Germany) for large correlation effects ($r \geq 0.50$) [18] and for the increase of $R^2$ in stepwise-forward regression analyses that failed to improve the simple models.

## 3. Results

Figure 2 shows means and standard deviations of jump height of all jump types for both sexes. Males (SJ: $0.38 \pm 0.06$; CMJ: $0.44 \pm 0.06$; CMJA: $0.50 \pm 0.04$; VSJ: $0.62 \pm 0.07$ [m]) jumped higher than females (SJ: $0.24 \pm 0.05$; CMJ: $0.26 \pm 0.04$; CMJA: $0.31 \pm 0.05$; VSJ: $0.36 \pm 0.08$ [m]) across all jump types ($F(1,1) = 1512.01$, $p < 0.001$, pη2 = 0.98). Two-way ANOVA with repeated measures revealed differences between jump types ($F(1,3) = 280.15$, $p < 0.001$, pη2 = 0.91) and an interaction between jump type and sex ($F(1,3) = 26.75$, $p < 0.001$, pη2 = 0.49). Pairwise comparison revealed differences in jump height for all pairs of jump types ($p < 0.001$).

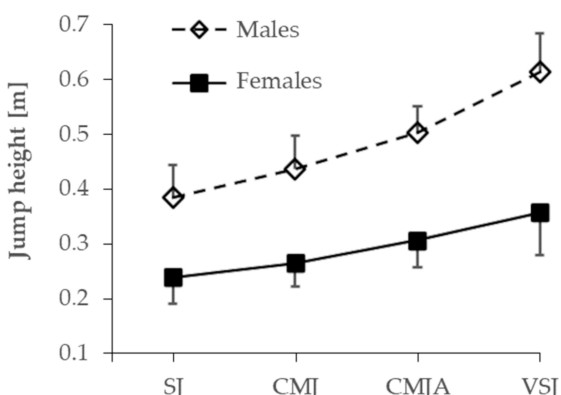

**Figure 2.** Mean and standard deviation of jump heights during SJ, CMJ, CMJA, and VSJ in females and males.

Figure 3 displays the distributions and correlations of jump height between general jumps and VSJ. In females, VSJ correlated with SJ ($r = 0.88$, $p < 0.001$), CMJ ($r = 0.88$, $p < 0.001$), and CMJA ($r = 0.82$, $p < 0.001$). In males, VSJ correlated with SJ ($r = 0.80$, $p < 0.001$), CMJ ($r = 0.85$, $p < 0.001$), and CMJA ($r = 0.75$, $p < 0.001$).

Correlation results between VSJ height and kinematic and kinetic characteristics during SJ, CMJ, and CMJA in females and males are presented in Table 1.

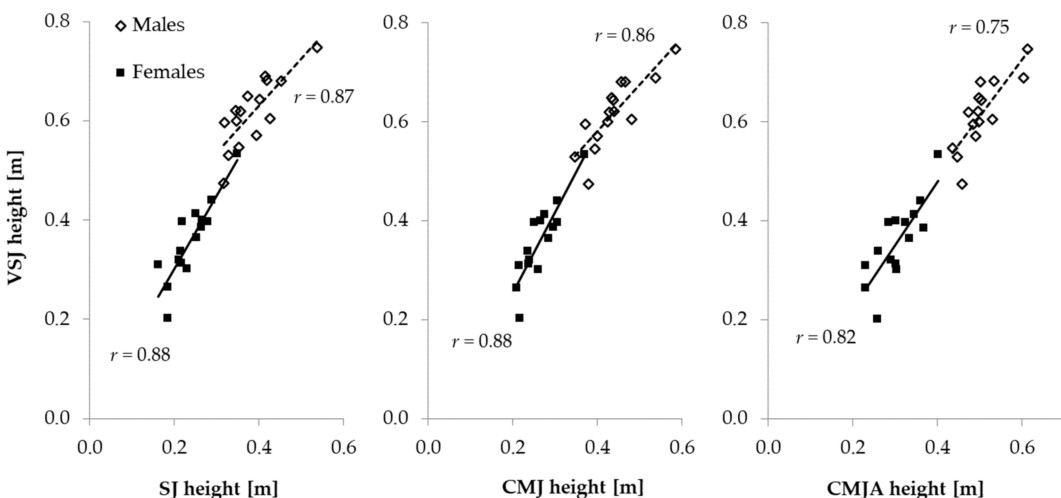

**Figure 3.** Each participant's mean jump heights of all jump types and correlation between VSJ and general jump types in females and males.

**Table 1.** Correlation results between volleyball spike jump height and variables during general jump types in females and males.

| | | **Females** | | | **Males** | | |
| --- | --- | --- | --- | --- | --- | --- | --- |
| | | **SJ** | **CMJ** | **CMJA** | **SJ** | **CMJ** | **CMJA** |
| Jump height | $r=$ | 0.88 | 0.88 | 0.83 | 0.80 | 0.85 | 0.75 |
| | $p=$ | *** | *** | *** | *** | *** | ** |
| Impulse | $r=$ | 0.40 | 0.32 | 0.56 | 0.50 | 0.26 | −0.65 |
| | $p=$ | 0.07 | 0.13 | * | * | 0.17 | ** |
| Minimal ND knee angle | $r=$ | 0.26 | −0.52 | −0.28 | −0.15 | −0.08 | 0.04 |
| | $p=$ | 0.17 | * | 0.17 | 0.30 | 0.39 | 0.45 |
| Minimal D knee angle | $r=$ | 0.28 | −0.47 | −0.26 | −0.10 | −0.05 | 0.04 |
| | $p=$ | 0.16 | * | 0.18 | 0.37 | 0.43 | 0.45 |
| CoM decrease | $r=$ | −0.08 | 0.41 | 0.27 | 0.10 | 0.03 | 0.06 |
| | $p=$ | 0.40 | 0.07 | 0.18 | 0.37 | 0.46 | 0.43 |
| Maximal RFD FP1+2 | $r=$ | 0.75 | 0.05 | 0.12 | 0.65 | −0.13 | −0.39 |
| | $p=$ | ** | 0.44 | 0.34 | ** | 0.32 | 0.09 |
| Maximal RFD FP1 | $r=$ | 0.78 | 0.15 | 0.13 | −0.05 | −0.13 | −0.50 |
| | $p=$ | *** | 0.30 | 0.33 | 0.43 | 0.32 | * |
| Maximal RFD PF2 | $r=$ | 0.70 | −0.01 | 0.06 | 0.47 | −0.08 | −0.27 |
| | $p=$ | ** | 0.48 | 0.42 | * | 0.39 | 0.18 |
| Torso incline | $r=$ | −0.21 | −0.18 | −0.29 | 0.46 | −0.23 | 0.19 |
| | $p=$ | 0.23 | 0.26 | 0.16 | * | 0.21 | 0.26 |
| Maximal angular velocity ND knee | $r=$ | 0.70 | 0.69 | 0.60 | 0.53 | 0.54 | 0.21 |
| | $p=$ | ** | ** | * | * | * | 0.24 |
| Maximal angular velocity D knee | $r=$ | 0.75 | 0.80 | 0.72 | 0.73 | 0.55 | 0.37 |
| | $p=$ | ** | *** | ** | ** | * | 0.10 |
| Maximal angular velocity ND shoulder | $r=$ | | | 0.14 | | | 0.17 |
| | $p=$ | | | 0.31 | | | 0.29 |
| Maximal angular velocity D shoulder | $r=$ | | | 0.06 | | | 0.05 |
| | $p=$ | | | 0.41 | | | 0.43 |

Note: SJ = squat jump; CMJ = countermovement jump; CMJA = countermovement jump with arm swing; D = dominant; ND = non-dominant; RFD = rate of force development; FP = force plate; * < 0.05; ** < 0.01; *** < 0.001.

Regression predictions of VSJ height based on general jump types were improved from the simple model (i.e., jump height of general jump types as the only predictor) in females for SJ from $R^2 = 0.76$, $p < 0.001$ to $R^2 = 0.92$, $p < 0.05$ and in males for SJ from $R^2 = 0.61$, $p < 0.01$ to $R^2 = 0.71$, $p < 0.05$ and for CMJA from $R^2 = 0.52$, $p < 0.01$ to $R^2 = 78$, $p < 0.01$. The equations of the improved regression models were as follows:

Female VSJ height = $-0.09 + 1.48 \cdot$SJ height + $1.78 \cdot 10^{-4} \cdot$SJ maximal RFD FP2 − $7.47 \cdot 10^{-5} \cdot$SJ maximal RFD FP1 and FP2,　(1)

$$\text{Male VSJ height} = 0.23 + 0.65 \cdot \text{SJ height} + 2.93 \cdot 10^{-5} \cdot \text{SJ maximal RFD FP2}, \quad (2)$$

$$\text{Male VSJ height} = 0.37 + 0.96 \cdot \text{CMJA height} - 1.23 \cdot 10^{-4} \cdot \text{CMJA impulse}. \quad (3)$$

No improvements were found for the simple models of CMJ ($R^2 = 0.76$, $p < 0.001$) and CMJA ($R^2 = 0.65$, $p < 0.001$) in females and CMJ in males ($R^2 = 0.69$, $p < 0.001$).

Statistical power to obtain improved models was at $1 - \beta = 0.53$ for CMJ in females, $1 - \beta = 0.22$ for CMJA in females, and $1 - \beta = 0.33$ for CMJ in males. Statistical power to detect correlation effects. Statistical power to detect correlation effects $r \geq 0.50$ was at $1 - \beta = 0.89$.

## 4. Discussion

A main objective of this study was to investigate the suitability of using general jump types to assess sport-specific VSJ performance. Correlations were found between VSJ height and all measured general jump heights (i.e., SJ, CMJ, and CMJA). This suggests that general jump types may be suitable for training and testing to improve and assess VSJ jump height. However, regression analyses revealed that only 52% to 76% of the variance in VSJ height could be explained in models using general jump height as the only predictor. The observation that VSJ height differed from the height of all other jump types raises doubts regarding if general jump heights alone are sufficient to assess VSJ performance. Predictions with a precision of 52% in the weakest model are not recommended, and the VSJ performance should not be decided prematurely on the basis of general jump height alone.

Differences in jump height were also found between sexes during all four jump types. Jump performance differences between sexes were previously reported in volleyball players during SJ [19], CMJ [19], and VSJ [6]. The reasons may include biological and power differences [20,21], the ability to benefit from stretch-shortening-cycles [19,22], and technical-coordinative variations [6]. These sex differences support sex-specific assessment of jump performances. Moreover, ANOVA results revealed an interaction (sex by jump type) for jump height, indicating that differences in height between jump types vary for females and males. Therefore, sex-specific analyses of jump performance are highly recommended, especially for predictions of VSJ height based on general jump type performances.

Simple models with only one predictor (i.e., SJ, CMJ, or CMJA height) achieved $R^2$ values between 0.52 and 0.76. Considering that these results are an equivalent of the explained variance in the criterion variable (i.e., VSJ height), it seems reasonable not to draw conclusions on sport-specific VSJ performance during the testing of general jump types. This should be considered especially when testing CMJA in males, where the least favorable result was achieved. General jump testing has several benefits (e.g., standardization, comparability, reliability) and is frequently used for performance testing in various sports [10]. The findings of the current study allowed improved association of general jump performance with VSJ favoring improved prediction. Improved regression models were derived from SJ performance in both sexes and CMJA performance in males. These models improved considerably the predictions of VSJ height from $R^2 = 0.76$ to $R^2 = 0.92$ in SJ for females, from $R^2 = 0.61$ to $R^2 = 71\%$ in SJ for males, and from $R^2 = 0.52$ to $R^2 = 0.78$ in CMJA for males. This suggests improvements for explaining the variance in the model by 26%, 10%, and 26%, respectively, and none of the improved models required more than two variables added to the respective general jump height.

For practical application of the current findings, it is important to note the strength of all three improved models requiring only kinetic variables as additional predictors. No kinematic data were required except for the jump height of the respective general jump type. Since the jump height of all mentioned general jump types could be calculated via ground reaction forces, the measurement of kinetic data is sufficient for the application of the predication equations presented in this study and, thus, an advanced assessment of jump performance. A limitation of this study is that only three of six regression models could be improved. The other three analyses achieved statistical power ranging between

$1 - \beta = 0.22$ and $1 - \beta = 0.53$. Considering the achieved power for correlation analyses (i.e., $1 - \beta = 0.89$) and the correlation effects of the added kinematic and kinetic variables, the power in failed model attempts was not caused by sample size. Instead, it can be reasoned by weak partial correlations with VSJ height (with simple jump type height as control variable). The authors selected the investigated variables based on their best knowledge and the current state of research. To improve future optimization of models, the authors recommend identifying predictors that correlate stronger with VSJ jump height while showing no co-linearity with the height in simple jump types. The presented data suggest that this is more promising than increasing the sample size.

The overall best results for each sex were derived by the improved models during SJ in females ($R^2 = 0.92$) and CMJA in males ($R^2 = 0.78$). Therefore, these are the recommended jump types during performance testing to allow the best predictions of VSJ height if applying the improved models. If the application of improved models is not feasible requiring the use of a single-predictor assessment, the current data suggest the use of SJ and CMJ for females (both: $R^2 = 0.76$) and CMJ for males ($R^2 = 0.69$). VSJ did not correlate any better with CMJA than with CMJ, and CMJ was not superior over SJ in both sexes. This may be surprising as it could be expected that jump types that share more similarities would result in greater correlation. SJ; however, achieved better correlation with VSJ height than CMJA. An explanation may be that SJ reflected the capability to reduce the degree of muscle slack and enhance the buildup of muscle [23], allowing for faster power generation that is essential in explosive movements like volleyball jumps [8,24–26]. CMJ and CMJA can provide information about the qualities of stretch-shortening-cycle during the countermovement [19], which is an important aspect in all volleyball jumps and VSJ performance [27]. Therefore, measuring both jump types (with and without countermovement) during performance testing can provide differential information about two abilities that are both relevant in volleyball jumping.

## 5. Conclusions

Jump height achieved during commonly used general jump types (i.e., SJ, CMJ, and CMJA) correlated well with VSJ height. However, regression predictions of VSJ height based on general jump height as the only predictor were not sufficiently accurate. In addition, differences between sexes for all jump types and an interaction (sex by jump types) suggested the need for additional sex-specific variables to improve the association of general jump performance with VSJ height. Depending on the model, one to two additional kinetic variables were sufficient to greatly improve regression prediction of VSJ height based on general jump performance. Improved models were found for SJ in females and in SJ and CMJA in males. In females, SJ, and in males, CMJA improvements were especially noteworthy (i.e., +26% of the variance explained). The current study provides the derived regression equations to apply the improved models in jump performance testing. These models provide well-established and reliable general jump types that may be used for testing while providing more precise and valuable information about the sport-specific VSJ performance.

**Supplementary Materials:** The following are available online at https://www.mdpi.com/2076-341 7/11/3/1105/s1, Data S1: data sheet, containing the data used in this study.

**Author Contributions:** Conceptualization, P.X.F. and H.W.; methodology, P.X.F., J.M. and H.W.; formal analysis, P.X.F. and J.M.; investigation, P.X.F., J.M., H.-J.K.M., S.P.v.D. and H.W.; resources, D.H. and H.W.; writing—original draft preparation, P.X.F. and J.M.; writing—review and editing, P.X.F., J.M., D.H., H.-J.K.M., J.W.B., S.P.v.D. and H.W.; visualization, P.X.F. and J.M.; supervision, H.W.; project administration, H.W. All authors have read and agreed to the published version of the manuscript.

**Funding:** This research received no external funding.

**Informed Consent Statement:** Informed consent was obtained from all subjects involved in the study.

**Acknowledgments:** We wish to thank the players and the coaches of the PSV Salzburg for participation in this study.

**Conflicts of Interest:** The authors declare no conflict of interest.

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
