# Peer review of "Relationship between General Jump Types and Spike Jump Performance in Elite Female and Male Volleyball Players"

_applsci, doi:10.3390/app11031105_

Round 1

Reviewer 1 Report

I do not understand the sense of the study

why to predict  an easy jump ? What's the point of predicting a jump that's very, easy to measure?
and on top of that complicate it with cinematic variables for me and feeling it a lot the work is not acceptable in these terms

please,review the research question and objectives

Author Response

A: We thank the reviewer for the effort and the overall positive assessment of our manuscript. Regarding the reviewer's comment (see below), there must have been a misunderstanding, which we clarify in our answer. This was no issue for the other reviewers.

R: I do not understand the sense of the study

why to predict  an easy jump ? What's the point of predicting a jump that's very, easy to measure?
and on top of that complicate it with cinematic variables for me and feeling it a lot the work is not acceptable in these terms

please,review the research question and objectives

A: The study did not predict an “easy jump”. Instead, it predicted the complex jump height. This prediction was based on general jumps, which could be called easy jumps. This was also the purpose and value of this study. The study provides a new salutation that allows using simple jumps for the assessment of a complex jump that cannot be tested as easily. The objectives were clearly stated in the last paragraph of the introduction.

Reviewer 2 Report

Very interesting topic. Your paper was well organized but did have a few grammatical error.

Abstract: what kinematic variables improve the predictive models

Introduction: well written 

Materials and Methods:

participants: well written 

study design:

was the warm up consistent between subjects?

why was a one minute break selected and was it long enough?

Data collection: well written

Variables and definitions:well written 

Statistical analysis: well written

Results: Table 1 is very busy and difficult to follow can to make it into two tables

Discussion: are there any other variables or test that were not used  (single leg vertical jump for example) that may improve the modesl

Conclusion: well written

Author Response

A: We thank the reviewer for the well-balanced comments and the overall positive feedback. We implemented the reviewer's suggestions to strengthen the manuscript.

Very interesting topic. Your paper was well organized but did have a few grammatical error.

A: The manuscript was reviewed by two native speakers who publish in the sport scientific field for many years.

Abstract: what kinematic variables improve the predictive models

A: As per reviewer’s suggestion, the two variables were specified and named in the abstract.

Introduction: well written 

Materials and Methods:

participants: well written 

study design:

was the warm up consistent between subjects?

A: We added a clarification under 2.2 Study Design and Test Protocol. The warm-up was not strictly instructed but it was supervised to ensure that all athletes perform movements to warm-up the required parts of the body (as specified in the manuscript) to the individual extent that they feel comfortable and ready for the task.

why was a one minute break selected and was it long enough?

A: The 1-minute break was chosen to allow recovery, and the decision for 1 minute was supported by previous experiences in jump testing. We are sure that it was long enough because the athletes rarely asked for the full minute for recovery. Furthermore, statistical analyses showed great reliability between jumps within subjects. So, we cannot expect any decrease of performance due to potential fatigue effects.

Data collection: well written

Variables and definitions:well written 

Statistical analysis: well written

Results: Table 1 is very busy and difficult to follow can to make it into two tables

A: We think that Table 1 is not too crowded. However, if the editors agree with the reviewer that Table 1 in its current form is too crowded, we prepared and annexed an alternative option (Table 1A and Table 1B) at the end of the manuscript.

Discussion: are there any other variables or test that were not used  (single leg vertical jump for example) that may improve the modesl

A: We included the most promising tests and variables derived from the literature review. All these tests and variables are described in the manuscript, and the list of variables used for this study was deemed complete after comprehensive literature review. No other ones were included that were not reported in the manuscript. We cannot speculate about potential effects of other variables or tests.

Conclusion: well written

Reviewer 3 Report

The article takes up a very interesting issues. The authors cite 27 references but only 8 of them have been published last ten years. A literature review on this issue could be researched better. The research tools and the method of results analysis were selected correctly (see the detailed comments below). Some doubts are raised by the small size of the research group (see the comments below). However the research groups are comparable (in terms of numbers). In the "Discussion" chapter authors should indicate the limitations and strengths of the work. The reviewed paper contains a several publishing errors described in the detailed notes.

Detailed comments: 

  • in the "Materials and Methods" chapter: the order of the subchapters should be changed, should be: 2.2. Data Collection and Processing, 2.3. Study Design and Test Protocol;
  • in the „Data Collection and Processing” chapter: no photo / diagram of the test stand and no diagram of the location of markers on the body of the tested person

  • in the “Statistical Analysis” chapter: no information if the authors examined the power of the test; in case of the statistical analysis of the small group, the power of the test should be examined.

Author Response

A: We thank the reviewer for the comments that helped strengthening our manuscript.

The article takes up a very interesting issues. The authors cite 27 references but only 8 of them have been published last ten years. A literature review on this issue could be researched better.

A: Ten references were published within the last ten years (since 2010). Each reference was selected after careful literature review and found suitable to support the specific statements. Some older references present methodological guidelines or findings that still contribute and shape the current state of research. To our best knowledge and after comprehensive literature review, the selected references were found most suitable. If the reviewer disagrees, we kindly ask to specify any critical references.

The research tools and the method of results analysis were selected correctly (see the detailed comments below). Some doubts are raised by the small size of the research group (see the comments below). However the research groups are comparable (in terms of numbers). In the "Discussion" chapter authors should indicate the limitations and strengths of the work. The reviewed paper contains a several publishing errors described in the detailed notes.

A: Comments below and “publishing errors” were addressed below. The fourth paragraph of the Discussion chapter includes strengths of the study. As per reviewer’s suggestion, we reworded this section for clarification and we added limitations at the end of the paragraph.

Detailed comments: 

  • in the "Materials and Methods" chapter: the order of the subchapters should be changed, should be: 2.2. Data Collection and Processing, 2.3. Study Design and Test Protocol;

A: We prefer the current order. The reason is that study design and test protocol are always developed first and afterwards data will be collected and processed. If the reviewer and editors disagree, we follow the editor’s decision.

  • in the „Data Collection and Processing” chapter: no photo / diagram of the test stand and no diagram of the location of markers on the body of the tested person

A: The marker set and locations were specified in the second sentence of Data Collection and Processing, providing the reference with details. For better visualization, we added a figure of the markers as they were attached on a tested person.

  • in the “Statistical Analysis” chapter: no information if the authors examined the power of the test; in case of the statistical analysis of the small group, the power of the test should be examined.

A: We agree and added power analyses of the relevant data accordingly to 2.5. Statistical Analyses, 3. Results, and 4. Discussion.

Round 2

Reviewer 1 Report

dear authors did not answer the question raised.
main question:
why estimate the jump, when it can be easily measured ?
I'm sorry, but I don't understand the rationale  of the study.

Kind regards

Author Response

Reviewer: dear authors did not answer the question raised.
main question:
why estimate the jump, when it can be easily measured ?
I'm sorry, but I don't understand the rationale of the study.

Answer:

Dear reviewer, we are glad to answer your question. The predicted jump (i.e. VSJ) canNOT be measured easily and reliably. The reason is the difficulty for standardization due to the complexity of the movement (explanation and supporting references are provided in the third paragraph of the introduction section).

However, the VSJ is the jump of interest in the sport of volleyball. That is why, for the prediction of VSJ which canNOT be measured easily and reliably, our method uses jumps that CAN be measured easily and reliably. These are less complex jumps, which are very very frequently used in research, testing, and practice despite their lesser relevance compared to volleyball-specific performance. The reason is easy standardization, comparability, and great reliability of the less complex jumps (also this is explained and supported by references in the third paragraph of the introduction section).

Therefore, the rationale was to provide a prediction of VSJ which canNOT be tested easily and reliably, on the basis of jumps that are commonly used in testing because of their reliability.

To assess the hereby explained difficulty of testing beyond the references we used in the manuscript which support all our statements, please briefly consult the current literature to find that almost all studies in the field used the suggested less complex jumps for performance testing for volleyball-specific performance studies, despite VSJ performance being the topic of interest in these studies. We cannot add a full literature review on specifically this issue as this would exceed the length of any acceptable introduction, but all these explanations and statements are provided and supported by references in the submitted version of the manuscript.

Round 3

Reviewer 1 Report

all questions are responded satisfactorallly